# Role of Macrophages in Sickle Cell Disease Erythrophagocytosis and Erythropoiesis

**DOI:** 10.3390/ijms24076333

**Published:** 2023-03-28

**Authors:** Renata Sesti-Costa, Fernando F. Costa, Nicola Conran

**Affiliations:** Hematology and Hemotherapy Center, University of Campinas, UNICAMP, Campinas 13083-878, Brazilferreira@unicamp.br (F.F.C.)

**Keywords:** erythroblastic islands, hemolysis, iron, leukocytes, phagocytosis, red blood cells

## Abstract

Sickle cell disease (SCD) is an inherited blood disorder caused by a β-globin gene point mutation that results in the production of sickle hemoglobin that polymerizes upon deoxygenation, causing the sickling of red blood cells (RBCs). RBC deformation initiates a sequence of events leading to multiple complications, such as hemolytic anemia, vaso-occlusion, chronic inflammation, and tissue damage. Macrophages participate in extravascular hemolysis by removing damaged RBCs, hence preventing the release of free hemoglobin and heme, and triggering inflammation. Upon erythrophagocytosis, macrophages metabolize RBC-derived hemoglobin, activating mechanisms responsible for recycling iron, which is then used for the generation of new RBCs to try to compensate for anemia. In the bone marrow, macrophages can create specialized niches, known as erythroblastic islands (EBIs), which regulate erythropoiesis. Anemia and inflammation present in SCD may trigger mechanisms of stress erythropoiesis, intensifying RBC generation by expanding the number of EBIs in the bone marrow and creating new ones in extramedullary sites. In the current review, we discuss the distinct mechanisms that could induce stress erythropoiesis in SCD, potentially shifting the macrophage phenotype to an inflammatory profile, and changing their supporting role necessary for the proliferation and differentiation of erythroid cells in the disease. The knowledge of the soluble factors, cell surface and intracellular molecules expressed by EBI macrophages that contribute to begin and end the RBC’s lifespan, as well as the understanding of their signaling pathways in SCD, may reveal potential targets to control the pathophysiology of the disease.

## 1. Introduction

A single point mutation in the gene that encodes the beta globin chain results in the production of abnormal sickle hemoglobin (HbS) and sickle cell disease (SCD) [1]. SCD is encountered as homozygous sickle cell disease (HbSS), termed sickle cell anemia (SCA), or in compound heterozygosity with another hemoglobin gene mutation (e.g., HbC disease or beta thalassemia) [2], where the polymerization of HbS, when deoxygenated, and the consequent formation of hemoglobin fibers causes the sickling of red blood cells (RBCs), which is the primary cause of SCD pathophysiology [3]. RBC sickling incurs multiple disease mechanisms that ultimately trigger vaso-occlusion in blood vessels and end-organ damage [4]. The clinical complications of SCD are many and varied, and they can be classified as chronic (e.g., chronic pain, chronic kidney disease, sickle retinopathy, avascular necrosis) or acute (e.g., painful vaso-occlusive episodes, acute chest syndrome, stroke) in nature [4,5]. Of all the clinical complications of SCD, painful vaso-occlusive episodes and anemia, caused principally by continuous hemolysis, are considered the most characteristic of SCD [6].

## 2. Sickle Cell Disease Pathophysiology

While SCD is primarily a red cell disease, its pathophysiology is extremely complex [7]. The vaso-occlusive processes that obstruct the small blood vessels, particularly venules, culminate from inflammatory processes that drive the recruitment and adhesion of blood cells (activated leukocytes and platelets, as well as RBCs) to activated endothelial cells of the vascular wall, with the eventual trapping of sickled RBCs in heterocellular agglomerates and the interruption of blood flow [1]. The inflammatory processes that play a fundamental role in SCD pathophysiology are generated by multiple mechanisms [8]. Firstly, vaso-occlusive processes themselves are extremely inflammatory; the ischemia of blood vessels followed by their reperfusion when blood flow is restored leads to the release of damage-associated molecular patterns (DAMPs), such as extracellular heat shock proteins, circulating DNA, high mobility group box B1 (HMGB1) and interleukin (IL)-1α, as well as the generation of reactive oxygen species (ROS) [9,10]. Secondly, alterations in the physiochemical properties of sickle RBCs and repeated cycles of cell sickling make these cells extremely fragile, and red cell destruction (or hemolysis) can occur both extravascularly, in organs, and intravascularly [7]. Intravascular hemolysis is an extremely inflammatory event, whereby the release of hemoglobin and other potent DAMPs (such as ATP) into the circulation can trigger innate immune responses and pan-cellular activation [11]. Other alterations that contribute to inflammation in SCD include abnormalities at the membrane of the RBCs themselves (increased phosphatidylserine exposure and adhesion molecule activity, for example), thromboinflammatory processes, oxidative stress, and leukocytosis [8].

As well as initiating and propagating vaso-occlusive processes, inflammatory processes cause many of the chronic and acute complications of SCD as well as the chronic organ damage that is recognized as a worrying feature of SCD now that patients have a longer life expectancy [12,13]. Complications of SCD that are associated with inflammation include pulmonary hypertension, stroke, and acute chest syndrome [6,14,15].

## 3. The Role of Macrophages in Hemolysis Product Clearance

Hemolysis describes the premature destruction of RBCs (that usually have a lifespan of approximately 120 days) within the circulation or in tissues [16]. Under normal conditions, hemolysis of damaged or senescent erythrocytes occurs primarily extravascularly. The spleen filters the blood and, since its vascular architecture makes it extremely sensitive to erythrocytic alterations, this organ is usually the principal site of extravascular hemolysis, although clearance of RBCs by the liver and bone marrow can also occur [17]. Macrophages play a key protective role in the clearance erythrocyte contents, where the phagocytosis of altered RBCs in tissues by macrophages prevents the extracellular release of inflammatory hemoglobin and heme, and the consequent triggering of sterile inflammatory processes. Macrophages involved in erythrophagocytosis, such as the splenic red pulp macrophages, have a close relationship with and are situated close to the location of the RBCs. These macrophages express molecules that enable them to screen and consequently identify and phagocytize damaged or senescent RBCs; such molecules include sirp-α, which provides an inhibitory signal to the macrophage when it encounters CD47 on the surface of healthy RBCs [18]. Erythrophagocytosis increases intracellular heme concentrations in the macrophages, leading to the transformation of these cells into erythrophagocytes for iron recycling, in association with the triggering of antioxidative and anti-inflammatory pathways to defend against hemolytic stress [19] (Figure 1). The phenotypic transformation of erythrophagocytes into cells that down-regulate inflammatory processes is regulated by the transcription factors BACH1, SPI-C, and NRF2 [16]. Endogenous heme binds to the transcription factor SPI-C, releasing it from the repression of BACH1. Free SPI-C up-regulates the expression of genes involved in iron metabolism, such as heme oxygenase 1 (HMOX-1), which is the enzyme responsible for heme catabolism. In a murine model of spherocytosis, which is associated with extensive extravascular hemolysis, macrophages in the liver can phagocytose damaged erythrocytes to prevent the toxic effects of cell-free hemoglobin and its heme group, promoting the adoption of an anti-inflammatory phenotype [20]. Metabolic adaptation to heme detoxification in macrophages has been found to require a shift to the pentose phosphate pathway that is induced by heme-derived carbon monoxide [21].

Macrophages comprise a cell population with distinct phenotypes and functions dependent on their tissue location and also on the mediators present in the microenvironment, which can polarize them to a profile that fluctuates between M1 (or classically activated) and M2 (alternatively activated) macrophages. M1 macrophages are pro-inflammatory cells that are activated and polarized by pathogen-associated molecular patterns and by inflammatory mediators, whereas M2 macrophages are anti-inflammatory cells, generally polarized by IL-4 and IL-13, that are able to repair the tissue after damage [22]. Anti-inflammatory erythrophagocyte transformation can be reproduced in vitro by exposing mouse and human bone marrow macrophages to heme–albumin complexes, yielding a distinctive transcriptional signature that differentiates heme-polarized macrophages from M1- and M2-polarized cells, indicating a link between erythrocyte homeostasis and innate immunity [20]. Furthermore, recent studies employing single-cell RNA-sequencing with directional RNA velocity analysis found that heme-activated signaling by NRF2 caused the differentiation of murine bone marrow macrophages toward antioxidant, iron-recycling macrophages, at the expense of homeostatic dendritic cell generation, indicating that hemolytic stress may contribute to hyposplenism-related secondary immunodeficiency [23].

In SCD, congestion of the spleen due to the presence of less deformable RBCs results in continuous microinfarction of the venous sinuses, often causing splenic dysfunction and atrophy or splenomegaly during childhood with loss of splenic filtration capacity [24,25]. This skews the hemolytic process to occur predominantly intravascularly, with the release of cell-free hemoglobin and erythrocyte contents into the circulation rather than their controlled clearance by the phagocytic process (Figure 1). The hemoglobin that is released from RBCs during intravascular hemolysis can be cleared by scavenger proteins, such as haptoglobin (HP), produced mainly by hepatocytes, and further heme processing can occur via the binding of heme to hemopexin [26,27]. HP binds to hemoglobin tightly and prevents the oxidative effects and tissue extravasation of free hemoglobin. The liver, spleen and other tissues can take up the hemoglobin/HP complex principally by the CD163 receptor on macrophages and Kupffer cells [27,28]. During events characterized by excessive intravascular hemolysis, as may occur in SCD, efficient hemoglobin clearance mechanisms are overwhelmed, and HP and hemopexin depletion occurs [29,30,31,32]. Non-bound cell-free hemoglobin reacts with vascular nitric oxide (NO), incurring a significant reduction in local NO bioavailability and the generation of (Hb-Fe^3+^), amongst other products [33,34]. Since NO is also important for maintaining cellular homeostasis, a reduction in its vascular bioavailability during intravascular hemolysis can potentially augment vasoconstriction and bring about endothelial dysfunction and platelet activation amongst other effects [35,36]. The oxidized hemoglobin product, Hb-Fe^3+^, accumulates in the circulation and tissues [34], where it can release heme.

A number of heme-scavenging proteins exist, but the plasma glycoprotein, hemopexin (Hpx; which is produced by the liver), has the highest affinity for binding free heme [37]. Hemopexin–heme complexes are cleared from the organism by macrophagic endocytosis, mainly in the liver and spleen, via the Hpx receptor (CD91), with consequent degradation by the HMOX-1 enzyme [38,39]. Heme is a highly hydrophobic molecule that exerts multiple inflammatory effects, activating leukocytes, up-regulating cellular adhesion molecule and cytokine expression, and augmenting lipid peroxidation and oxidant production in its free non-bound form [40,41,42,43,44]. Free heme also induces the secretion of TNF-α via a macrophage-dependent Toll-like receptor 4 (TLR4) mechanism, and it induces NLRP3 inflammasome formation with the subsequent processing of IL-1β in macrophages and endothelial cells [45,46]. 

As such, increased intravascular hemolysis in SCD leads to excess vascular cell-free hemoglobin and heme, the depletion of scavenger proteins and reduced receptor-mediated hemoglobin/heme clearance. This reduction in erythrophagocytosis and controlled RBC product clearance, and the elevation in circulating free heme, triggers innate immune pathways and the shifting of the phenotype of macrophages to a pro-inflammatory phenotype (Figure 1). This shift in macrophage phenotype potentially influences erythropoiesis in SCD.

## 4. Leukocytosis and Sickle Cell Disease

SCD pathophysiology has long been associated with leukocytosis, even in the absence of infection [47]; among the factors that show statistically significant correlation with disease severity of SCD is steady-state neutrophil count, and an elevated baseline leukocyte count is associated with an increased risk of early death in patients [48,49]. Leukocyte activation in SCD plays a significant role in the initiation of vaso-occlusive events; this is due to the adhesion of more adherent leukocytes to the endothelium, their formation of heterocellular aggregates, neutrophil extracellular trap release and their contribution to inflammatory molecule production [50,51,52,53]. The mechanism by which leukocyte counts are increased in patients with SCD is unclear, but infections, elevated circulating GM-CSF levels [54] and, possibly, increased gut permeability leading to bacterial intestinal translocation [55], may contribute, at least in part, to high leukocyte numbers in SCD; furthermore, genetic variants at the HbF-modifier loci may also modulate leukocyte numbers [56]. One of the benefits of the use of hydroxyurea therapy for treating patients with SCD is suggested to be the rapid reduction in leukocyte counts that often occurs before fetal hemoglobin level induction [57].

Monocyte counts have been reported as significantly elevated in patients with SCA compared with healthy individuals, and indeed, they are reduced by hydroxyurea therapy [54]. Depletion of the non-classical patrolling monocyte subset (CD14^dim^CD16^+^) has been reported in SCD [58], which is possibly due to their modulation into inflammatory monocytes or their destruction following the scavenging of endothelial-adherent sickle RBCs [59]. Although heme-induced splenic macrophage expansion together with dendritic cell depletion has been reported in mice with SCD [23], as well as Nrf2-dependent macrophage hypercellularity during stress erythropoiesis (a feature of SCD) [60,61], it is not clear whether leukocytosis and the changes in monocyte profiles described in SCD contribute to drive the differentiation of macrophages and their role in erythropoiesis.

## 5. Stress Erythropoiesis in Sickle Cell Disease

Erythropoiesis is a complex and regulated process of generating about new 200 billion RBCs every day. Through the extremely high proliferation rate, each proerythroblast, the earliest erythroid-committed progenitor, is capable of generating 16 mature RBC at the end of the process. As development follows, the proliferative rate and the cell size reduce, whereas chromatin progressively condensates and hemoglobin synthesis is increased, differentiating the cells to basophilic, polychromatophilic and orthochromatic erythroblasts in sequence. The nucleus, most organelles and RNA content are extruded, generating the reticulocytes followed by mature RBCs that circulate in the bloodstream for about 120 days. Steady-state erythropoiesis occurs in specialized niches in the bone marrow called erythroblastic islands (EBIs), consisting of a central macrophage surrounded by erythroid cells at several stages of differentiation. Macrophages are responsible for providing a supportive microenvironment for the proliferation, survival, and differentiation of erythroid cells through direct cell–cell interactions and the secretion of growth factors [62,63]. Anemic conditions together with the chronic inflammation present in SCD activate mechanisms of stress erythropoiesis, thereby intensifying erythrocyte production by expanding the number of EBIs in and outside the bone marrow, mainly in the spleen and liver [64].

Tissue hypoxia induced by hemolytic anemia stimulates the production and systemic secretion of erythropoietin (Epo). The Epo receptor (EpoR) is expressed by both erythroid progenitors and macrophages and, when activated by Epo, it increases erythropoiesis through signaling pathways that differ between these cells. In erythroid cells, Epo up-regulates anti-apoptotic genes, ensuring their survival, and it increases the commitment of early progenitors to the erythroid linage. On the other hand, Epo also expands the number of macrophages and elevates their expression of adhesion molecules, generating new EBI niches via the induction of prostaglandin E_2_ [65]. The depletion of Epo/STAT5 signaling specifically in macrophages has been shown to inhibit stress erythropoiesis, supporting a fundamental role for macrophages in erythropoiesis during non-physiological situations [65].

Additionally, upon stress and following Epo stimulation, splenic red pulp macrophages and hepatic Kupffer cells secrete the chemokine CCL2, which recruits CCR2^+^ circulating monocytes to the respective organs, where they differentiate into EBI macrophages. CCR2 KO mice showed a drop in monocyte recruitment to the spleen and liver and a consequent reduction in erythroid cell expansion [66,67], showing that the new EBI structures that are formed during stress are critical for supporting extramedullary erythropoiesis, at least in murine models.

As described earlier, intravascular hemolysis in SCD and the release of free hemoglobin and heme into the bloodstream leads to continuous leukocyte activation and a chronic inflammatory state. Pro-inflammatory cytokines reduce steady-state erythroid production by skewing hematopoiesis toward the development of myeloid cells and inducing apoptosis in erythroid-committed cells [68,69]. In addition, pro-inflammatory mediators are frequently involved in the proliferation of cell progenitors, whereas pro-resolving cytokines stimulate the differentiation stage of the cells. Thus, chronic inflammation in SCD contributes to anemia by reducing the terminal development of erythroid cells [70]. Furthermore, inflammatory stimuli increase hepcidin production, which impairs iron exportation through ferroportin. This event stimulates erythrophagocytosis, further reducing the numbers of RBCs, characterizing anemia induced by inflammation [64,71]. As a compensatory pathway, pro-inflammatory cytokines stimulate a mechanism of stress erythropoiesis, inducing the proliferation of extramedullary erythroblasts [72].

Inflammation-induced stress erythropoiesis is activated by a distinct mechanism from Epo- and steady state-induced erythropoiesis. In mice models, the spleen is the primary site for erythroid development upon stress, although the liver becomes the site of erythrocyte production in splenectomized mice [73]. In steady-state erythropoiesis, CD34^+^ hematopoietic stem cells can potentially generate all cell lineages. Upon stress erythropoiesis, these stem cells leave the bone marrow, as illustrated by higher number of circulating CD34^+^ cells in SCD patients, and they migrate to the spleen or liver, where the BMP4/Smad5 signaling pathway is stimulated, committing cell fate to the erythroid linage [74,75].

Although the proliferation and expansion of CD34^+^ cells are driven by stem cell factor (SCF) in both steady-state and stress erythropoiesis, the latter is specifically dependent on the activation of growth and differentiation factor 15 (GDF15) and Wnt signaling. GDF15 induces BMP4 expression on macrophages by a HIF2α-dependent mechanism, which acts in erythroid progenitors, stimulating their proliferation. Accordingly, GDF15 KO mice have reduced expansion of splenic EBI and consequent inhibition of stress erythroblasts proliferation [76]. The following transition to erythroid linage is dependent on the role played by Epo in both progenitors and in macrophages. Epo/STAT signaling on macrophages represses Wnt signaling, shifting the developmental stage of erythroid cells from expansion to terminal differentiation [65].

In contrast to the steady-state scenario, in response to anemia, there is a rise in γ-globin expression, replacing β-globin in erythroid cells and generating fetal hemoglobin (HbF) [77], suggesting that stress erythropoiesis is involved in the origin of HbF^+^ cells in SCD. Indeed, the presence of macrophages in erythroid cultures demonstrated their influence in the shifting from β-globin to γ-globin expression in early stages of erythropoiesis, although the mechanisms involved are still obscure [78]. While HbF is implicated in disease improvement by reducing cell sickling, HbF is also capable of partially rescuing erythroblast death due to ineffective erythropoiesis [79]. In fact, over the past years, evidence has corroborated the hypothesis that hemolysis due to the lower membrane deformability of mature RBCs is not the only primary cause of anemia in SCD. It can also be impacted by a defect in the developmental stages of erythrocytes, as exemplified by the sickling of late erythroblasts in hypoxic conditions [80], such as those occurring in the bone marrow environment, and the elevated rate of cell death at the transition between polychromatic and orthochromatic stages of differentiation [79], showing the existence of ineffective erythropoiesis in SCD. The apoptosis of late erythroblasts also takes place under normoxic condition, although in lower numbers, proving the existence of additional mechanisms involved in ineffective erythropoiesis [81]. Thus, in addition to the immense amount of mature RBCs death, around 40% of polychromatic erythroblasts die due to ineffective erythropoiesis in SCD. It is not known whether the macrophages that perform the clearance of this elevated extra number of cells are the same as those forming the EBIs; such modulations in phenotype could influence their regular function in erythropoiesis, especially in the spleen and liver where macrophages play this new role in stress. Consistent with this mechanism, SCD patients have nucleated erythroid cells in their bloodstream [82], indicating the premature release of these cells from the EBI and suggesting a defect in macrophage–erythroblast interactions.

## 6. Macrophage–Erythrocyte Interactions in Sickle Cell Disease Erythropoiesis

Erythroid production is directly correlated to EBI number; thus, the EBI microenvironment is crucial for events of erythropoiesis. Macrophages present in these structures play several key regulatory roles in the process via both contact- and soluble factor-dependent mechanisms. These cells are able to induce erythroblast proliferation and survival, provide iron for hemoglobin synthesis, and engulf the nucleus expelled during erythroblast enucleation [83,84,85].

Although EBI macrophages are not a homogeneous population, they display an M2-like phenotype, with the expression of CD206 and CD169, and they exhibit an anti-inflammatory profile [83,86]. These anti-inflammatory macrophages express CD163, which scavenges the hemoglobin–haptoglobin complex besides acting as an adhesion receptor to erythroblasts, although its ligand is still not known [87]. The island’s integrity is achieved by the attachment of macrophages to erythroblasts, which is mediated by several adhesion molecules, such as VCAM-1, α_V_ integrin, ephrin-B2, and Emp expressed by EBI macrophages, which in turn bind to α4β1 integrin, ICAM-4, EphB4, and Emp, expressed by erythroblasts, respectively. Each molecule of these sets of receptors plays a role in the EBI structure, as demonstrated by mice models. The pharmacological inhibition of the interaction between either α4β1 integrin with VCAM-1 or ephrin-B2 with EphB4, as well as the depletion of ICAM-4, in mice results in lower numbers of EBI and the dissociation of erythroblasts from macrophages [88,89,90]; furthermore, the depletion of Emp is reportedly lethal due to severe anemia in mice fetuses [91]. Additionally, a secreted form of ICAM-4, ICAM-4S, was found to be released by orthochromatic erythroblasts and reticulocytes; this secreted molecule competes with the membrane-bound molecule by binding to α_V_ integrin on macrophages [92], potentially weakening reticulocyte–macrophage interactions and releasing them from the EBI niche toward the sinusoids.

These molecule sets seem to be heterogeneously distributed among EBI macrophages throughout the bone marrow [93], which might represent phenotypic and functional differences of EBIs located at distinct zones. For instance, macrophages scattered through the bone marrow are adhered to proerythroblasts and early erythroblasts, whereas macrophages close to sinusoids are attached to cells in the late stages of differentiation, such as polychromatic and orthochromatic erythroblasts [94]. It is not known, though, whether the macrophages are either different cells or cells in a distinct stage of maturation that migrate toward sinusoids upon erythropoiesis progression. Furthermore, some adhesion molecules have distinct levels of contributions to steady-state and stress erythropoiesis, indicating heterogeneity in the expression of surface molecules between EBI macrophages from physiological (bone marrow) and stress (spleen and liver) tissues. Mutations in the α4 integrin gene inhibit stress erythropoiesis, as illustrated by the impairment of recovery after anemia induced by hemolysis, but these mutations have little effect on steady-state erythropoiesis [95]. Additionally, it has been shown that the hematopoietic niche of SCD patients is extremely altered, with changes in the number and frequency of different cell progenitors and with modified EBI structure [96], which is possibly due to changes in adhesion protein expression.

It has been demonstrated that circulating monocytes, particularly the intermediate CD14^high^CD16^+^ cells, are able to boost the differentiation of CD34^+^ cells to erythrocytes in vitro in a contact-independent manner, increasing expansion and reducing the death of CD34^+^ cells before they enter the erythroblast stage. Like EBI macrophages, these monocytes express CD206, CD163 and CD169, enabling them to interact with erythroid cells and, in the presence of glucocorticoid, promote erythropoiesis and engulf the nucleus that is expelled during the erythroid differentiation process [86,97]. In SCD, where the number of circulating monocytes is elevated, these cells might play an important role in stress erythropoiesis since the spleen is frequently dysfunctional. In fact, patrolling monocytes from SCD patients were seen to phagocyte circulating RBCs [98]. Whether the iron recycled by those monocytes can be used for the synthesis of hemoglobin by the circulating erythroblasts present in SCD patients is a question to be investigated.

Between the orthochromatic erythroblast and reticulocyte stages of differentiation, erythroid cells expel the nucleus and most of their organelles, which are then cleared by EBI macrophages, promoting erythroid maturation. Indeed, the depletion of EBIs macrophages in mice impaired the elimination of mitochondria, resulting in reduced mature RBCs [99]. The extruded nuclei were shown to be recognized by their externalized phosphatidylserine (PS) and reduction in PS recognition, as occurs in MFG-E8 mutant mice that display impaired clearance of nuclei from erythroblasts [84]. Notably, the retention of mitochondria has been detected in both reticulocytes and RBCs in SCD patients [100]. The mitochondria in mature RBCs have been shown to be metabolically competent and were associated with high levels of ROS in SCD. In addition, mitochondria-containing RBCs are able to activate immune responses in vitro, indicating that the retention of mitochondria may play a role in the complications of SCD [101]. Together with the presence of circulating nucleated RBCs, the retention of mitochondria in RBCs suggests a defect in the elimination process by EBI macrophages in addition to the weakness in the macrophage–erythroblast interaction. The digestion of the nuclei and mitochondrial DNA collected from erythroid cells is essential for the survival of EBI macrophages and is performed by DNase II, whose absence in the KO mice model causes the death of the macrophages due to the accumulation of intracellular non-degraded DNA, resulting in lethal impairment in erythropoiesis and anemia [102].

Erythropoiesis is supported by the iron provision of EBI macrophages for hemoglobin synthesis. Macrophages recycle iron from erythrophagocytosis and the clearance of hemoglobin/Hp and heme/Hpx complexes through HMOX-1 activity, which is up-regulated in SCD and catabolizes heme to equimolar amounts of iron, biliverdin and carbon monoxide (CO). The role of HMOX-1 in stress erythropoiesis is essential, as a drop in its activity impairs the maturation of erythroid cells [103,104]. It has been demonstrated that besides iron, CO can support RBC maturation, at least in cultures of the erythroid cell lineage K562 [105]. Recycled iron can have one of two destinations, either stored as ferritin or exported by ferroportin and transported bound to transferrin (Trf) [106]. Erythroblasts can internalize the iron/Trf complex through their Trf receptor (CD71). In addition, macrophages have been reported to secrete ferritin in macrophage/erythroblast cocultures, which is then captured by erythroblasts [107]. The systemic control of iron load in SCD is complex, as hepcidin is secreted in response to iron and inflammation, whereas it is down-regulated by hypoxia and erythropoiesis. Additionally, recent findings have shown that despite anemic conditions, treatment with a ferroportin inhibitor improved some aspects of SCD pathology by restricting iron to RBCs and consequently reducing the HbS concentration [108]. Thus, a better understanding of iron metabolism, specifically in the EBI niche, is still needed and would provide insights for novel therapeutic approaches.

KLF1 (Kruppel Like Factor 1) is the transcription factor that best defines EBI macrophages. It is responsible for up-regulation of DNase II and the expression of VCAM-1, CD163, CD169, enabling the macrophage to promote erythroid development [109,110,111]. In the absence of KLF1 expression, mice have reduced DNase II and die of severe anemia [112,113]. KLF1 also regulates the expression of IL-33, which is a cytokine that promotes the maturation of RBCs [111]. Additionally, IL-33 differentiates monocytes into macrophages involved in erythrophagocytosis and iron recycling [114] and has been shown to be elevated in SCD patients [115]. Other soluble factors found to be secreted by EBI macrophages are insulin-like growth factor (IGF1) and IL-18 [109]. IGF1 was shown to increase erythropoiesis by binding to the IGF1R expressed by erythroblasts in several development stages [116], whereas the possible role of IL-18 in erythropoiesis has still not been investigated, even though the expression of IL-18R has been detected in erythroid lineage cells and the production of IL-18 may be stimulated in SCD by heme-activated inflammasome formation [117]. Although the presence of systemic levels of these factors in SCD has been investigated, the contribution of EBI macrophages to the production and the dynamics of these mechanisms specifically in the EBI niche in SCD patients still needs clarification.

Depletion of CD169^+^ macrophages in mice models of polycythemia vera and β-thalassemia, which both display activated stress erythropoiesis, was able to reverse some aspects of the diseases pathology, such as the defect in erythroid development and lifespan as well as splenomegaly [83,118], revealing EBI macrophages as possible targets for future therapies for stress and ineffective erythropoiesis in disorders such as SCD. The secreted factors induced in the EBI niche should be investigated as an option to enhance the proliferation and maturation of RBCs, whereas the signaling molecules that are modulated upon stress erythropoiesis are a potential target for pharmaceutical approaches. Furthermore, uncovering the mechanisms by which EBI macrophages influence the switch to HbF production in erythroid cells would shed light on other options for HbF-inducing treatments to improve the clinical manifestations of SCD patients.

## 7. Concluding Remarks

Isolating and characterizing EBI macrophages in human tissues is a challenge; much of our understanding of stress erythropoiesis comes from mouse models, which may not reflect entirely what happens in humans. Furthermore, although the role of macrophages in promoting erythropoiesis is well recognized, the mechanisms by which macrophages influence RBC development and the role that macrophages play in the stress erythropoiesis that is encountered in SCD are only now becoming more evident. Indeed, recent evidence of the heterogeneity of EBI macrophages has revealed that adult murine bone marrow EBI may also provide a site for terminal granulopoiesis, which may be favored over erythropoiesis during anemia of inflammation [119]. Due to the existence of an extremely inflammatory environment in SCD, it would be important to examine the balance between EBI-driven erythropoiesis and myelopoiesis in SCD as well as the macrophage’s role in such an equilibrium.

Extensive intravascular hemolysis and splenic dysfunction likely drive macrophages to a more inflammatory phenotype as opposed to their more protective RBC clearance/iron recycling phenotype. These inflammatory macrophages intensify the production of pro-inflammatory molecules in SCD with potentially important consequences, including further monocyte proliferation, and macrophage migration and expansion at sites of erythropoiesis. Inflammatory molecules, together with hypoxic processes, modulate the function of the expanded macrophage population within the EBI to accelerate erythroid cell proliferation but in a manner that promotes the apoptosis of erythroblasts and their premature release from the EBI (Figure 2) while elevating the erythroid content of beneficial fetal hemoglobin. As such, understanding how phenotypic changes in macrophages in SCD contribute to modulate erythropoiesis, and EBI function, as well as hinder iron recycling, will be essential for finding ways to ameliorate erythropoiesis efficiency in a disease characterized by anemia.

## Figures and Tables

**Figure 1 ijms-24-06333-f001:**
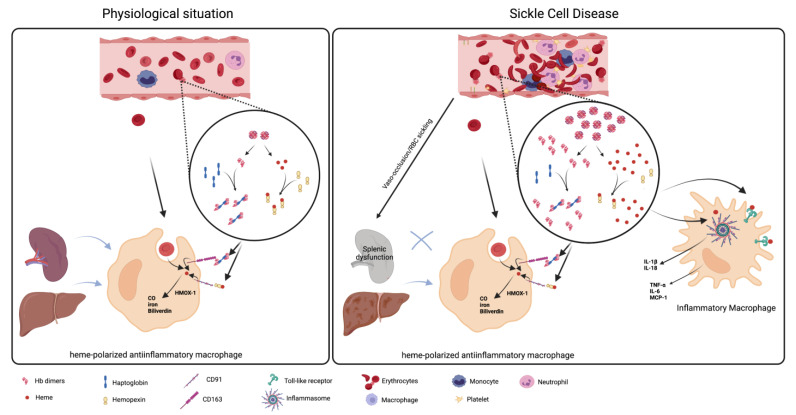
**Macrophage clearance of red blood cells and hemoglobin products in sickle cell disease.** Under normal physiological conditions, damaged or senescent erythrocytes are erythrophagocytosed extravascularly, primarily in the spleen. The elevation of intracellular heme during erythrophagocytosis transforms the cells into erythrophagocytes that process hemoglobin and recycle iron, preventing the damaging effects of these molecules. Scavenger proteins, such as haptoglobin and hemopexin, which bind cell-free hemoglobin and heme molecules, respectively, counter physiological levels of red blood cell (RBC) lysis in the vasculature and their release of cell-free RBC products. Haptoglobin–hemoglobin and hemopexin–heme complexes bind to the CD163 and CD91 receptors, respectively, on macrophages, particularly in the liver, where these molecules are broken down by heme oxygenase-1 (HMOX-1) into iron, carbon monoxide (CO), and biliverdin. These heme-polarized macrophages have an anti-inflammatory phenotype that differs from those of classic M1/M2-polarized macrophages. In sickle cell disease, vaso-occlusive processes and congestion of the spleen causes splenic dysfunction, reducing the erythrophagocytosis of senescent RBCs. In association with the skewing of hemolysis to occur intravascularly, hemoglobin polymerization and RBC sickling significantly augment RBC damage and the lysis of these cells in the circulation. Considerable elevations in cell-free hemoglobin and heme saturate hemoglobin clearance proteins, leading to the binding of heme to pattern recognition receptors, such as Toll-like receptors, on macrophage membranes and the entry of unbound heme into these cells. Innate immune pathways, including inflammasome formation, are triggered and macrophages take on an inflammatory phenotype, resulting in inflammatory molecule production and pyroptosis. Created with Biorender.

**Figure 2 ijms-24-06333-f002:**
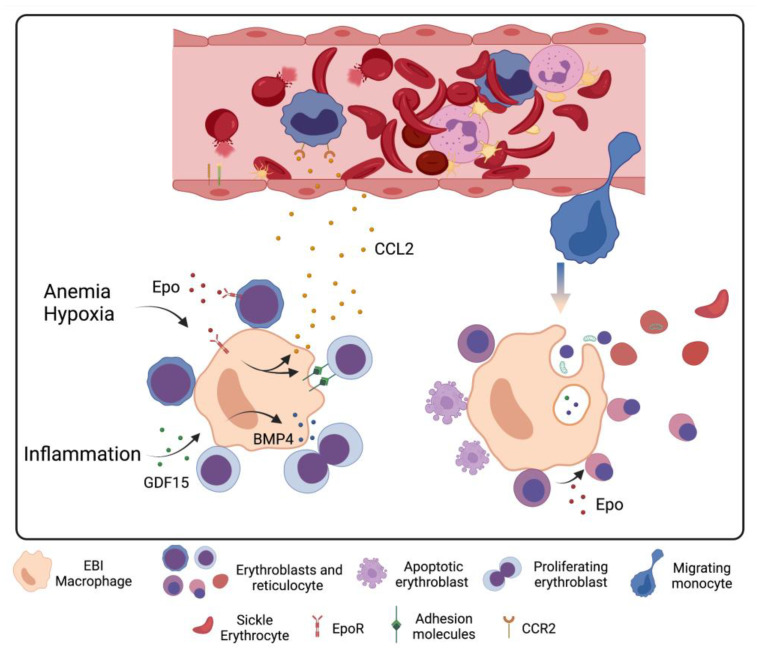
**Role of macrophages in stress erythropoiesis in sickle cell disease.** Upon homeostasis, macrophages located in the EBI of the bone marrow provide support for erythroblast proliferation and development through production of soluble factors and adhesion molecules, iron provision and engulfment of the extruded nucleus and organelles. Under conditions of anemia and hypoxia, macrophages sense the increased level of erythropoietin (Epo) through the EpoR, which elevates the set of adhesion molecules responsible for the attachment to different development stages of erythroblasts. The Epo–EpoR interaction also up-regulates the production of CCL2, which attracts circulating CCR2-expressing monocytes to bone marrow and to extramedullary sites, especially spleen and liver, where they differentiate in EBI macrophages, increasing erythropoiesis. Inflammation, on the other hand, stimulates erythropoiesis by a distinct stress-specific mechanism. Inflammatory cytokines induce the production of BMP4 by EBI macrophages through the elevation of GDF15. BMP4 stimulates the proliferation of erythroblasts attached to macrophages, increasing stress erythropoiesis. The production of HbS by late erythroblasts in SCD is implicated in their sickling, which causes an excessive rate of cell death between the polychromatic and orthochromatic stages of differentiation, resulting in an ineffective erythropoiesis. Created with Biorender.

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
