# Peer review of "Role of Macrophages in Sickle Cell Disease Erythrophagocytosis and Erythropoiesis"

_ijms, 2023, doi:10.3390/ijms24076333_

Round 1

Reviewer 1 Report

Sesti-Costa et al., are submitting a review on the role of macrophages in sickle cell disease erythropoiesis.

As clearly described by the authors, sickle cell disease (SCD) is more than a simply red cell disorder. The pathophysiology involves several cell types and many cellular and molecular mechanisms are impaired in non-erythroid cells. Hemolysis and inflammation are 2 major interconnected processes both initiating and promoting SCD main complications.

Among the non-erythroid cell type involved in SCD pathophysiology, the macrophage has emerged as one of the targets of hemolysis and inflammation through its role in erythrocyte clearance.

Thus, the aim of the current article is to review the role of macrophage in SCD erythropoiesis.

The review is very well written and easy to follow. The figures are very helpful and didactic. The current article is gathering a lot of information and data on the role of macrophages in erythropoiesis but also more generally on the role of macrophages in RBC clearance in SCD. Thus, the title should be modified and not restricted to erythropoiesis.

Studying EBI is not easy as bone marrow is not always accessible and managing to isolate human intact EBI is a real technical challenge. Thus, most of the current study have been performed in mice. This is an important point that should be discussed in the conclusion. This is very important as extra medullar erythropoiesis is occurring in mice.

Minor comments:

-The authors did well introduce SCD and the major complications. I would remove the paragraph on the therapeutics that is not related directly to the macrophages.

-Macrophages include several subtypes and thus are involved in several functions. Thus, a paragraph should be dedicated to the description of these different populations. It would be helpful to better understand what the M1- and M2-polarized cells are (line 110).

-Not all the macrophages are specialized in erythrophagocytosis. The authors should define more precisely these specialized macrophages.

-line 101: BACH1, SPI-C, NRF2 and ATF1 are transcription factors involved in the transformation of erythrophagocytes into antiinflammatory erythrophagocytes. Could the authors develop this idea? What are the roles of these transcription factors? What do the authors mean by transformation? Does the phenotype of these macrophages change (surface markers, secreted proteins,…)?  

-Figure 1: the authors mentioned CD163 and CD91. Are these molecules expressed on all macrophage subtypes? CD163+CD91+ macrophages are found mainly in the liver, not the spleen, that is the main site of erythrophagocytosis?

-line 174-175: the authors explained that the saturation of the erythrophagocytosis process leads to increase cell free-heme and hemoglobin ultimately resulting in the shifting of erythrophagocyte to a pro-inflammatory phenotype and that is this change could affect erythropoiesis. The hypothesis is very attractive but it feels hard to understand how the phenotype shifting of erythrophagocyte mainly located in spleen and liver would impact another type of highly specialized macrophages in another localization i.e. the bone marrow.

-As for macrophages involved in erythrophagocytose, EBI macropahges should be better described.

-line 233: The two studies mentioned are very interesting but the authors should mentioned that extra medullar erythropoiesis in mice is physiological but is not, or at a lesser extent, in human.

-line 290: the release of nucleated cells in the bloodstream strongly suggests a defect in the tightly regulated process of red blood cell production. More recently, it has also been shown that mature circulating red blood cells contains mitochondria in SCD patients. Could the authors comment on these new finding?

-line 300: the authors stated that EBI macrophages express CD206 and CD169 and are anti-inflammatory macrophages. Is the expression of the 2 proteins different from spleen/or liver macrophages? A table summarizing the main surface markers of the several macrophages subtypes would more helpful.

-line 300: the authors mentioned that EBI macrophages are heterogenous. Could the authors be more explicit? Are EBI macrophages from the bone marrow different from EBI macrophages from spleen or liver?

-line 406: the sentence should be rewritten. Even if the involvement of macrophages in erythropoiesis is undeniable, pretending that it is extremely clear is not accurate as a lot of questions regarding the interactions, the communications between the macrophages and the erythroid progenitors remain to be clarified.

- A recent paper suggests that EBI regulate granulopoiesis in parallel to terminal erythropoiesis (Romano et al., Blood, 2022). What is the opinion of the authors on this publication?

The authors did write an very complete and nicely written review that emphasize the complexity of SCD.

Author Response

We thank the reviewers and the Editor for their important and relevant comments about our manuscript. We feel the suggested additions certainly improved the quality of the review. All the reviewers’ comments were carefully addressed in our point-by-point reply below and in the revised manuscript as tracked changes. Thus, the title of the manuscript was changed, and the references were adjusted to align the information inserted or removed as suggested. We hope that this new version is suitable for publication in International Journal of Molecular Sciences.

Reviewer’s comments:

Reviewer #1:

Sesti-Costa et al., are submitting a review on the role of macrophages in sickle cell disease erythropoiesis.  

As clearly described by the authors, sickle cell disease (SCD) is more than a simply red cell disorder. The pathophysiology involves several cell types and many cellular and molecular mechanisms are impaired in non-erythroid cells. Hemolysis and inflammation are 2 major interconnected processes both initiating and promoting SCD main complications.

Among the non-erythroid cell type involved in SCD pathophysiology, the macrophage has emerged as one of the targets of hemolysis and inflammation through its role in erythrocyte clearance.

Thus, the aim of the current article is to review the role of macrophage in SCD erythropoiesis.

The review is very well written and easy to follow. The figures are very helpful and didactic. The current article is gathering a lot of information and data on the role of macrophages in erythropoiesis but also more generally on the role of macrophages in RBC clearance in SCD. Thus, the title should be modified and not restricted to erythropoiesis.

Reply: We thank the Reviewer for their comments about the manuscript. We have changed the title to include the role of macrophages in RBCs clearance, as suggested. The new title is “Role of Macrophages in Sickle Cell Disease Erythrophagocytosis and Erythropoiesis”

Studying EBI is not easy as bone marrow is not always accessible and managing to isolate human intact EBI is a real technical challenge. Thus, most of the current study have been performed in mice. This is an important point that should be discussed in the conclusion. This is very important as extra medullar erythropoiesis is occurring in mice.

Reply: Isolating and characterizing intact EBI in human bone marrow is indeed a challenge. Also, despite the evidence indicating that extramedullary erythropoiesis also occurs in humans in non-physiological situations, most of the knowledge built so far about this phenomenon comes from murine models, which may not reflect entirely what happens in humans. A statement acknowledging this fact was added to the concluding remarks as suggested. Please see Lines 431-35.

“Isolating and characterizing EBI macrophages in human tissues is a challenge; much of our understanding of stress erythropoiesis comes from mouse models, which may not reflect entirely what happens in humans. Furthermore, although the role of macrophages in promoting erythropoiesis is well recognized, the mechanisms by which macrophages influence RBC development and the role that macrophages play in the stress erythropoiesis that is encountered in SCD are only now becoming more evident.”

Minor comments:

-The authors did well introduce SCD and the major complications. I would remove the paragraph on the therapeutics that is not related directly to the macrophages.

Reply: We accepted the reviewer’s suggestion and removed “Sickle cell disease therapeutics” section and the related references.

-Macrophages include several subtypes and thus are involved in several functions. Thus, a paragraph should be dedicated to the description of these different populations. It would be helpful to better understand what the M1- and M2-polarized cells are (line 110).

Reply: A brief clarification on M1 and M2 macrophage functions and how they are polarized was included for a better understanding of the difference between these and heme-polarized macrophages.

Please see lines 116-122:

“Macrophages comprise a cell population with distinct phenotypes and functions dependent on their tissue location and also on the mediators present in the microenvironment, which can polarize them to a profile that fluctuates between M1 (or classically activated) and M2 (alternatively activated) macrophages. M1 macrophages are pro-inflammatory cells that are activated and polarized by pathogen-associated molecular patterns and by inflammatory mediators, whereas M2 macrophages are anti-inflammatory cells, generally polarized by IL-4 and IL-13, that are able to repair the tissue after damage [22]).”

-Not all the macrophages are specialized in erythrophagocytosis. The authors should define more precisely these specialized macrophages.

Reply: Additional information about macrophages that phagocyte RBCs was provided in “The Role of Macrophages in Hemolysis Product Clearance” section as follows (Lines 96-101): “Macrophages involved in erythrophagocytosis, such as in the splenic red pulp macrophages, have a close relationship with and are located close to the RBC’s location. These macrophages express molecules that enable them to screen, and consequently phagocyte damaged or senescent RBCs, such molecules include sirp-a, which provides an inhibitory signal to the macrophage when it encounters CD47 on the surface of healthy RBCs [18].”

-line 101: BACH1, SPI-C, NRF2 and ATF1 are transcription factors involved in the transformation of erythrophagocytes into antiinflammatory erythrophagocytes. Could the authors develop this idea? What are the roles of these transcription factors? What do the authors mean by transformation? Does the phenotype of these macrophages change (surface markers, secreted proteins,…)? 

Reply: To clarify, we meant that by recognizing and phagocyting RBCs or the heme-hemopexin and hemoglobin-haptoglobin complexes, macrophages are not classically activated, as opposed to free-heme activated macrophages. Instead, they are specialized in iron metabolism, which requires a shift in metabolic pathways. These heme-polarized macrophages show a distinctive transcriptional anti-inflammatory signature that differentiate them from M1- and M2-polarized cells. Internalized heme binds to the transcription factor SPI-C, releasing it from the repression of BACH1. Free SPI-C up-regulates the expression of genes involved in iron metabolism, such as heme oxygenase 1 (HMOX-1), which is the enzyme responsible for heme catabolism. With regard to NRF2, we mention in the text its involvement in the antiinflammatory phenotype of heme-polarized macrophage, as regulation of heme-induced inflammation was lost in NRF2-deficient mice. We added some of the information mentioned above to better clarify the role of the transcription factors in this process (Lines 107-110).

-Figure 1: the authors mentioned CD163 and CD91. Are these molecules expressed on all macrophage subtypes? CD163+CD91+ macrophages are found mainly in the liver, not the spleen, that is the main site of erythrophagocytosis?

Reply: The Reviewer raises an important point. The expression of CD163 and CD91 varies among macrophages from different tissues. According to Immgen.org data, these molecules are expressed by both splenic and liver macrophages, which indicates that both types of macrophage have the ability to clear heme-hemopexin and hemoglobin-haptoglobin complexes. We have acknowledged the role of macrophages in the spleen in the process in lines 249 and 265.

-line 174-175: the authors explained that the saturation of the erythrophagocytosis process leads to increase cell free-heme and hemoglobin ultimately resulting in the shifting of erythrophagocyte to a pro-inflammatory phenotype and that is this change could affect erythropoiesis. The hypothesis is very attractive but it feels hard to understand how the phenotype shifting of erythrophagocyte mainly located in spleen and liver would impact another type of highly specialized macrophages in another localization i.e. the bone marrow.

Reply: The intense intravascular hemolysis generates a high amount of free heme and hemoglobin which exceeds the capacity of hemopexin and haptoglobin to scavenge. Thus, instead of being cleared by macrophages through CD163 and CD91, free heme in the circulation activates TLR4 and the inflammasome, stimulating the secretion of inflammatory mediators. This is represented in Figure 1. One hypothesis is that EBI macrophages located either in the bone marrow or in the spleen and liver, in the case of stress erythropoiesis, can sense and be activated by free heme, which could affect their main function in erythropoiesis.

-As for macrophages involved in erythrophagocytose, EBI macropahges should be better described.

Reply: The phenotype of EBI macrophages described throughout the “Macrophage-Erythrocyte Interactions in Sickle Cell Disease Erythropoiesis” section, is similar to the phenotype of EBI macrophages in physiological situations. The data known so far that indicate differences between EBI macrophages in steady state versus in sickle cell disease are limited and were discussed in the section. In general, EBI macrophages express CD169, CD206, CD163, and the adhesion molecules VCAM-1, aV integrin, ephrin-B2, and Emp. They can phagocyte and digest the nucleus and organelles extruded by erythroblasts, and they are able to recycle iron that is used by the new erythrocytes in development. We have also discussed KLF1, the transcription factor responsible for the expression of most of the molecules mentioned above that characterize EBI macrophages.

-line 233: The two studies mentioned are very interesting but the authors should mentioned that extra medullar erythropoiesis in mice is physiological but is not, or at a lesser extent, in human.

Reply: We now acknowledge at the end of the paragraph that the data cited were obtained from murine models (line 255).

-line 290: the release of nucleated cells in the bloodstream strongly suggests a defect in the tightly regulated process of red blood cell production. More recently, it has also been shown that mature circulating red blood cells contains mitochondria in SCD patients. Could the authors comment on these new finding?

Reply: The Reviewer raises an important point, under physiological situations, erythroblasts eliminate their mitochondria during the terminal differentiation process. However, it was shown both in mice and humans that a percentage of RBCs retain mitochondria in SCD. The mitochondria in mature RBCs were shown to be metabolically competent and was associated with high levels of ROS in SCD. In addition, mitochondria-containing RBCs were able to activate immune response in vitro, indicating that retention in mitochondria can play a role in the complications of SCD. We have added the above information in the manuscript and another recent reference on this matter (Moriconi et al., 2022; PMID: 35670632), Lines 373-378.

-line 300: the authors stated that EBI macrophages express CD206 and CD169 and are anti-inflammatory macrophages. Is the expression of the 2 proteins different from spleen/or liver macrophages? A table summarizing the main surface markers of the several macrophages subtypes would more helpful.

Reply: Thank you for this suggestion. EBI macrophages from bone marrow, and also in the spleen and liver, express CD206 and CD169 upon stress erythropoiesis. However, the expression of surface markers in EBI macrophages is heterogeneous, even throughout the bone marrow, dependent on their stage and/or location, which makes it a technically difficult task to identify all the molecules expressed by EBI macrophages and their variation in quantity in different tissues or even in different locations of the same tissue. CD206 is also expressed by alternatively activated macrophages, which complicates their discrimination in non-physiological situations. Thus, a table with the main surface markers of macrophages may not represent them in an accurate way, at least with the knowledge we have so far.

-line 300: the authors mentioned that EBI macrophages are heterogenous. Could the authors be more explicit? Are EBI macrophages from the bone marrow different from EBI macrophages from spleen or liver?

Reply: EBI macrophages are heterogeneous throughout different locations inside the bone marrow, which is reflected by their adherence to distinct phases of erythroblasts, dependent on their location. EBI macrophages close to sinusoids are more specialized in digestion of the nuclei and organelles extruded by erythroblasts during the terminal stages of development and can also detach from these cells, releasing them into the circulation. In addition, data from experimental models show that some adhesion molecules that promote macrophage-erythroblast interactions have different contributions to physiological and stress erythropoiesis, which indicates that EBI macrophages from distinct tissues are also heterogeneous in their expression of surface molecules. We added a statement in the manuscript to clarify this idea in lines 344 - 347.

“Furthermore, some adhesion molecules have distinct levels of contributions to steady state and stress erythropoiesis, indicating heterogeneity in the expression of surface molecules between EBI macrophages from physiological (bone marrow) and stress (spleen and liver) tissues.”

-line 406: the sentence should be rewritten. Even if the involvement of macrophages in erythropoiesis is undeniable, pretending that it is extremely clear is not accurate as a lot of questions regarding the interactions, the communications between the macrophages and the erythroid progenitors remain to be clarified.

Reply: We agree that “extremely clear’ may provoke a wrong idea on that matter, and we have change the statement accordingly (lines 426-428).

- A recent paper suggests that EBI regulate granulopoiesis in parallel to terminal erythropoiesis (Romano et al., Blood, 2022). What is the opinion of the authors on this publication?

 Reply: Thank you for bringing this up. The paper mentioned brings very important information on the plasticity of EBI macrophages, which can adapt to generate erythrocytes or granulocytes (at least this has been demonstrated so far) dependent on the stimulation. This phenomenon has been shown in the bone marrow and it would be fundamental to understand whether it also occurs with EBI macrophages outside the bone marrow and especially in the context of SCD. We have inserted text into the concluding remarks regarding this very interesting recent finding.

Concluding Remarks – Lines 436-442:

“Indeed, recent evidence of the heterogeneity of EBI macrophages has revealed that adult murine bone marrow EBI may also provide a site for terminal granulopoiesis, which may be favored over erythropoiesis during anemia of inflammation [119]. Due to the existence of an extremely inflammatory environment in SCD, it would be important to examine the balance between EBI driven erythropoiesis and myelopoiesis in SCD, as well as the macrophage’s role in such an equilibrium.”

The authors did write an very complete and nicely written review that emphasize the complexity of SCD.

Reply: We thank the Reviewer for their positive comment on the manuscript, and for their extremely thorough review and very pertinent comments and suggestions.

Reviewer 2 Report

Sesti-Costa et al. explores the roles that macrophages play in sickle cell disease erythropoiesis. This review summarizes the events surrounding stress erythropoiesis process in SCD, such as the splenic inefficiency, intravascular hemolysis that causes anemia and hypoxia, and inflammatory activation of macrophages. Overall, the review is well written, comprehensive and accessible.

Minor comment:

In figure 1, haptoglobin is represented as blue dumbbell shaped molecules. Please correct the figure legend underneath to match the same color as the one used in the schematic.

Author Response

We thank Reviewer 2. We thank you for your very positive comments about our manuscript and for pointing out the error in figure 1. This has been corrected, thank you.